# Flavonoid/Polyphenol Ratio in *Mauritia flexuosa* and *Theobroma grandiflorum* as an Indicator of Effective Antioxidant Action

**DOI:** 10.3390/molecules26216431

**Published:** 2021-10-25

**Authors:** Juan Carlos Carmona-Hernandez, Mai Le, Ana María Idárraga-Mejía, Clara Helena González-Correa

**Affiliations:** 1Grupo de Investigación Médica, Línea Metabolismo-Nutrición-Polifenoles (MeNutrO), Universidad de Manizales, Manizales 17000, Colombia; jucaca@umanizales.edu.co; 2Research Group on Nutrition, Metabolism and Food Security, Universidad de Caldas, Manizales 17000, Colombia; anneidarraga@gmail.com; 3Institute of Pharmacy and Molecular Biotechnology, Faculty of Biosciences, Heidelberg University, 69120 Heidelberg, Germany; maianh.le96@yahoo.de

**Keywords:** polyphenols, flavonoids, antioxidant activity, *Mauritia flexuosa* (aguaje), *Theobroma grandiflorum* (copoazú)

## Abstract

Studies on polyphenols and flavonoids in natural products reveal benefits in the prevention of multiple diseases. Proper extraction, treatment of extracts, and quantification of polyphenols and flavonoids demand attention from the scientific community in order to report more specific biological action. Total polyphenol content (TPC) and total flavonoid content (TFC) (measured at three different times) of ethanol, methanol and acetone extracts of *Mauritia flexuosa* (aguaje) and *Theobroma grandiflorum* (copoazú) fresh pulp, from the Colombian Amazon region, were evaluated with the purpose of focusing in the polyphenol/flavonoid proportion and its effective antioxidant activity. This objective could help to explain specific flavonoid biological action based on higher flavonoid proportion rather than higher total polyphenol content. Differences in extracting solvents resulted in statistically significant different yields; the highest TPC was observed with acetone 70% in *Mauritia flexuosa* and ethanol 80% for *T. grandiflorum.* The best flavonoid/polyphenol ratio in *M. flexuosa* was about 1:2.4 and 1:12.8 in *T. grandiflorum* and the antioxidant efficacy was proportionally higher for flavonoids extracted from *T. grandiflorum*. HPLC analysis revealed 54 µg/g of the flavonoid kaempferol in *M. Flexuosa* and 29 µg/g in *T. grandiflorum*. Further studies evaluating this proportionality, in seeds or peel of fruits, as well as, other specific biological activities, could help to understand the detailed flavonoid action without focusing on the high total polyphenol content.

## 1. Introduction

Polyphenol compounds, and among these flavonoids in particular, make a substantial contribution to the antioxidant activity of specific fruits [1]. Studies on humans revealed an increased fat oxidation and improved insulin sensitivity in overweight or obese males by plant flavonoids [2,3,4]. Furthermore, combined histological and microarray analysis of aortas revealed local anti-proliferative and anti-inflammatory effects and a reduction of ovarian cancer cell viability [5,6]. Given multiple health benefits of polyphenols, it is of general interest to know the phenolic content and major phenolic compounds, such as flavonoids, found in regularly harvested and consumed fruits [7,8,9].

The most studied biological property in polyphenols is related to oxido-reducing activity. Oxidative stress is defined as an imbalance between oxidants and antioxidants in favour of the oxidants, leading to a disruption of redox signalling and control and/or molecular damage [10]. Growing investigations in this field implicate oxidative stress in the development of many diseases like cardiovascular diseases, neurodegenerative disorders or cancer [11,12]. Plants and fruits can contribute in the reduction of negative manifestations in multiples pathologies.

*Mauritia flexuosa* (Arecaceae) also known as buriti palm, is widely distributed in the Amazon region, in Colombia, Venezuela, the Guyanas, Trinidad, Ecuador, Peru, Brazil, and Bolivia [13]. It is characterized by an orange, soft, water-soluble edible pulp. Many ethnic groups in South America use it for a variety of medicinal treatments. For example, the oil extracted from the mesocarp is utilized to cure respiratory problems, pneumonia, influenza, snake bites and heart problems [14]. In Colombia it is commonly named “aguaje” and is represented in Figure 1.

Another commercial crop cultivated in the Amazon region is *Theobroma grandiflorum* (Malvaceae) known in Colombia as “copoazú” [16,17]. The yellowish-white, fleshy and insoluble in water, fruit pulp is eaten raw or prepared as juices, ice cream, sweets, and jams among other products [17]. The regular intake of this fruit and its derivatives might lead to a lower incidence of metabolic diseases. Studies have shown that phenolic-rich extracts of *Theobroma grandiflorum* decreased lipid peroxidation and increased plasma and tissue antioxidant capacities in high-fat fed rats [18,19]. Several studies have examined the total phenolic content in *T. grandiflorum* and *M. flexuosa*, but a comparison of the most effective extraction solvents (time of extraction) and the flavonoid/polyphenol ratio and its relation to antioxidant activity, to the best of our knowledge, has not been reported yet [20,21,22,23]. The objective of this study was to identify the best extraction approaches (solvent and time of extraction) leading to the highest yield of polyphenols and flavonoids and the consequent antioxidant activity in the Amazonian fruits *M. flexuosa* and *T. grandiflorum.*

## 2. Results and Discussion

### 2.1. Total Polyphenol Content (TPC)

TPC for *M. flexuosa* (aguaje) and *T. grandiflorum* (copoazú) are shown in Figure 2. Total polyphenols were extracted, from the edible part of both fruits, in 80% ethanol, 70% methanol and 70% acetone. Both pulps registered higher TPC levels in acetone with significant differences in comparison to polyphenol extracts in ethanol and methanol. The more water soluble fruit pulp, *M. flexuosa*, yielded better TPC registering more than 640 mg GAE/100 g in fresh pulp in 70% acetone. Previous studies, coinciding with the present work, reported higher TPC, for *M. flexuosa*, with values of 435.08 ± 6.97 and 362.90 ± 7.98 mg GAE/100 g of fresh pulp [23,24]. For *T. grandiflorum* studies revealed a total phenolic content of 40.3 mg GAE/100 g of fresh pulp [25]. In the present work the 70% acetone extract in total polyphenols from Colombian *T. grandiflorum* registered more than 50 GAE/100 g of fresh pulp. Table 1 shows accumulated TPC and TFC values in all extracts and at different extraction times for both fruits.

### 2.2. Total Flavonoid Content (TFC)

Similar to the TPC values, the flavonoid extraction process of *M. flexuosa*, TFC was significantly higher with 70% acetone. However, no differences were found with respect to extraction times. Total flavonoids in 80% ethanol and 70% methanol were similar showing significant differences within treatments and extraction times (24, 48 and 72 h). Considering TFC in *T. grandiflorum* extracts (as reported in Figure 2) the best results were obtained in 80% ethanol with no significant differences with respect to other solvents or extraction times.

According to other studies *M. flexuosa* reports the highest TPC and TFC values being comparable to results of the present study [22,23]. With respect to solvent affinity, 70% acetone yields better results than 80% ethanol and 70% methanol for the case of polyphenols and flavonoids in this highly water-soluble fruit pulp adding the contents at three different extraction times. With respect to polyphenols and flavonoids in *T. grandiflorum*, a very fleshy and water insoluble pulp, higher TPC was obtained in 70% acetone, and the best TFC was registered in 80% ethanol (added concentrations at three different times). Considering the evaluation of specific biological action of flavonoids, the proportion of flavonoids in the total polyphenol content could provide focused information. Table 1 reports the quantitative relation of flavonoids/polyphenols extracted from the pulp of *M. flexuosa* and *T. grandiflorum*. 

For the extraction of polyphenols and flavonoids in the fleshy (and water insoluble fruit pulp of *T. grandiflorum* the flavonoid:polyphenol registered, as shown in Table 1, very diluted presence of flavonoids with respect to the total polyphenol content being 80% ethanol the most efficient solvent. *M. flexuosa* and *T. grandiflorum* could serve as potential sources of natural antioxidants. Although higher TPC represents higher TFC values; the proportion of flavonoids in the group of total polyphenols is important to understand how a specific amount of flavonoids can provide more effective biological action. Flavonoids in the pulp of *T. grandiflorum* are 12 to almost 80 times more diluted than flavonoids in the pulp of *M. flexuosa* (with flavonoid/polyphenol ranging from 1:24 to 1:4.1).

### 2.3. Antioxidant Activity Based on 2,2-Diphenyl-1-picrylhydrazyl (DPPH) Radical Scavenging Assays

Total antioxidant compounds, extracted in 80% ethanol, 70% methanol and 70% acetone at 24, 48, and 72 h, in aguaje, as shown in Figure 3, yielded a total radical scavenging activity (inhibitory concentration (IC50%) when 50% of the antioxidant compounds are reduced) of 220.2 mg ascorbic acid equivalents per 100 g of fresh pulp (mg AAE/100 g FP) as a product of the addition of means in each extraction process. Antioxidant activity of polyphenols and flavonoids found in *T. grandiflorum* IC50% was 44.0 mg AAE/100 g FP. With respect to the proportionality for antioxidant compounds in each fruit pulp, phenolic compounds in *M. flexuosa* displayed five-fold greater antioxidant efficiency.

Antioxidant activity of polyphenols and flavonoids in aguaje was highest with significant differences in 70% acetone showing no differences with respect to extraction times and displaying a flavonoid:polyphenol proportion closer range (1:2.4 to 1:4.0). In the case of copoazú the proportions are wider (1:12 to 1:79) with respect to the presence of flavonoids in total polyphenol content, but DPPH action was proportionally closer lower with statistical differences with respect to extraction times. These results suggest that having even proportions of polyphenols and flavonoids in two very different fruit pulps, with respect to water solubility, antioxidant compounds in *T. grandiflorum* would display more efficient antioxidant activity.

### 2.4. HPLC/UV-Vis Spectrophotometry Flavonoid Identification

Results obtained by the performed analysis are presented in Figure 4. *M. flexuosa* shows a kaempferol content of 54.43 µg/g and for 29.04 µg/g for *T. frandiflorum*. Thus, the concentration of kaempferol in *M. flexuosa* extracts is 1.86-fold greater than in *T. grandiflorum*. Quercetin was identified in both fruits with concentrations below the limit of quantification. Bataglion et. al., reported 41.54 µg kaempferol/g and 83.27 quercetin/g of dry weight in the pulp of *M. Flexuosa* extracted with methanol. The higher amounts of quercetin could be due to the different solubility of the substance in water and methanol [26].

This HPLC data confirms the presence of flavonoids kaempferol and quercetin in the extracts from the present work. Limitations in the availability of more reference standards invite for posterior chromatographic evaluations. Other studies report the presence of more polyphenols and flavonoids in these fruits; more studies and different extraction methods for flavonoids from these fruits are recommended since combined histological and microarray analysis of aortas revealed local anti-proliferative and anti-inflammatory effects by quercetin [5,27]. Studies with kaempferol nanoparticles lead to a selective inhibition of ovarian cancer cell viability [6,28]. Thus, the investigated fruits could serve as potential sources of kaempferol and quercetin to be used in multiple purposes.

## 3. Materials and Methods

### 3.1. Reagents and Chemicals

Acetone, ethanol, sodium carbonate, and Folin-Ciocalteu reagent were purchased from PanReac AppliChem, ITW Reagents, (Darmstadt, Germany), acetonitrile and methanol, from Sigma Aldrich (St. Louis, MO, USA). Sodium nitrite, aluminium chloride obtained from LOBA Chemie (Mumbai, India), and sodium hydroxide from EMSURE Merck (Darmstadt, Germany). HPLC reference standards quercetin and kaempferol were acquired from MP Biomedicals (Invine, CA, USA). Ascorbic acid and 2,2-diphenyl-1-picrylhydrazyl (DPPH, Sigma-Aldrich) were purchased from Merck KGaA, (Darmstadt, Germany).

### 3.2. Plant Material and Sample Preparation

*T. grandiflorum* pulp and *M. flexuosa* pulp with seeds were purchased in Leticia (Colombian Amazon). The samples were refrigerated prior to all laboratory analyses. The extraction of phenolic compounds, in two different solvents, was performed according to the method described by Tauchen et al. [22], with modifications. Five grams of seedless samples were mixed with 25 mL ethanol 80%, methanol 70%, and acetone 70%, stirred for 30 min at 500 rpm, homogenized and stored at room temperature for 24h in the dark. The extracts were centrifuged for 10 min at 3500 rpm and the supernatant was recovered. Three different extractions times (24, 48, and 72 h) were tested in every extraction process and runs were done in triplicate.

### 3.3. Total Polyphenol Content (TPC)

The total polyphenol content of the (seedless) pulp extracts was determined using the Folin-Ciocalteu assay [29]. One millilitre of each extract was mixed with 1 mL Folin-Ciocalteu reagent (1:10), allowed to react 2 min, and was combined with 2 mL sodium carbonate (Na_2_CO_3_) (3.5% *w*/*v*). Reactants were incubated at room temperature for 90 min and protected from light. The absorbance was then read at 655 nm in a UV-Vis spectrophotometer (Mecasys Optizen POP, Daejeon, Korea). Results were calculated based on a gallic acid standard calibration curve. TPC is expressed as milligrams gallic acid equivalent (GAE) per 100 g of fresh pulp (mg GAE/100 g FP).

### 3.4. Total Flavonoid Content (TFC)

An aluminium chloride (AlCl_3_) colorimetric assay was the means used to quantify total flavonoids according to Phuyal et. al. [30] with modifications. Amounts of 0.5 mL of each extract, in triplicates, were added into test tubes with 2.0 mL of distilled water. To each solution 150 µL of sodium nitrite (NaNO_2_) (5%) was added followed by a 5-s vortex after 5 min 150 µL of AlCl_3_ (10%) were added to each solution following the same agitation and reaction time. Finally, 1 mL of sodium hydroxide (NaOH) (1M) and 1.2 mL of distilled water were added with vortexing and a reaction time of 5 min [30]. Total flavonoid content is reported as milligram quercetin equivalents (QE) per 100 g of fresh pulp (mg QE/100 g FP).

### 3.5. DPPH Assay for Radical Scavenging, Antioxidant, Activity

The DPPH radical scavenging assay is reported as the most useful technique for the search of the antioxidant activity in polyphenols and flavonoids from natural products. This DPPH compound is a stable free radical when prepared in methanol. The free radical scavenging assay was implemented to test antioxidant action of polyphenols from both fruits according to methods described by Shimu et al., [31] and Yeddes et al. [32] with slight modifications to optimize reactant volumes and reaction time. Volumes of 1900 μL of DPPH (100 μM) prepared in pure methanol were mixed with 100 μL of each diluted (1:5) extract and let react in the dark at room temperature for 30 min. The antioxidant activity of polyphenols and flavonoids from both fruit pulps was measured via spectrophotometry at 517 nm comparing against a methanol blank and using a positive control of ascorbic acid based on a calibration curve. The control curve was prepared with concentrations of comparable reference ascorbic acid (Merck KGaA) in a concentration range from 50 to 600 μg/mL. All dilutions followed the same DPPH reaction conditions for the antioxidant activity evaluated in fruit extracts. The slope taken from the calibration curve served for the calculation of the inhibition concentration (IC50%) when 50% of the antioxidant component is reduced. Results for IC50% were determined based on Equation (1):
% scavenging DPPH free radical = (ABS_Control_ − ABS_Extracts_/ABS_Control_) × 100%(1)

The antioxidant activity from polyphenols and flavonoids extracted from the pulp of *M. flexuosa* and *T. grandiflorum* is reported as mg of ascorbic acid equivalents per 100 g of fresh pulp (mg AAE/100 g FP).

### 3.6. HPLC/UV-Vis Spectrophotometry Flavonoid Identification

Pulp of both fruits was homogenized in a commercial blender in a proportion of 1:20 with distilled water. Homogenized solutions were centrifuged for 10 min at 3500 rpm. Supernatants were paper filtered (6 µm). HPLC analyses was performed in a Dionex UltiMate 3000 system (Thermo Fisher Scientific, Waltham, MA, USA) with a 250 mm × 4.6 mm i.d., 5 µm particle size column (Macherey-Nagel, Düren, Germany) in a binary mobile phase. Eluent A was HPLC grade water and eluent B was acetonitrile [33]. The elution gradient started from 40% acetonitrile and increased to 50% acetonitrile in 10 min at a flow rate of 1.5 mL/min. Flavonoids were analysed at 373 nm. External flavonoid standards for the characterization and quantification of kaempferol and quercetin were prepared in a mixture of methanol:ethanol (70% *v*/*v*, 1:1). Dilutions were done at eight different concentrations (1.0 to 100 mg/L) for each calibration curve. Flavonoid identification is reported as a comparison of retention times and quantitation is expressed as micrograms of each standard per gram of pulp extract (µg/g extract).

### 3.7. Statistical Analysis

All analyses were run in triplicates and are expressed as mean ± standard deviation (SD). Means were compared using the ANOVA test followed by Tukey test (*p* < 0.05) with the SPSS Statistics software version 20.0 (IBM, Armonk, NY, USA) with previous normality and homogeneity tests for all data sets.

## 4. Conclusions

The total phenolic and total flavonoid content in pulp extracts of *M. flexuosa* and *T. grandiflorum* using three different solvents (ethanol, methanol, and acetone) was evaluated at three different times (24, 48, and 72 h) in the present study. The results produced significant results with respect to solvent and time dependent efficacy for the extraction process. Our results, from statistical analyses, suggest that an extraction with 70% acetone yields higher contents and a better flavonoids/polyphenols ratio for *M. flexuosa* a highly water soluble fruit pulp. A different case was seen with *T. grandiflorum* which is a fleshy water insoluble fruit pulp and where the best antioxidant activity was detected in polyphenols and flavonoids extracted in 80% ethanol. The tested extracts showed efficient antioxidant activity with no direct proportionality with respect to higher content of total polyphenols and flavonoids suggesting that specific ratios (flavonoids:polyphenols) are important to consider the efficacy of antioxidant, or other, functionalities of extracted compounds.

## Figures and Tables

**Figure 1 molecules-26-06431-f001:**
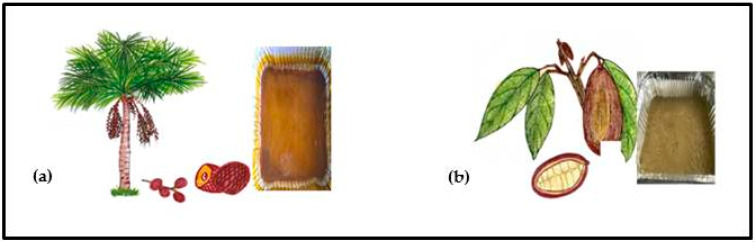
(**a**) *Mauritia flexuosa* (aguaje) palm, fruit [15], and seedless pulp; (**b**) *Theobroma grandiflorum* (copoazú) plant, fruit [15], and seedless pulp.

**Figure 2 molecules-26-06431-f002:**
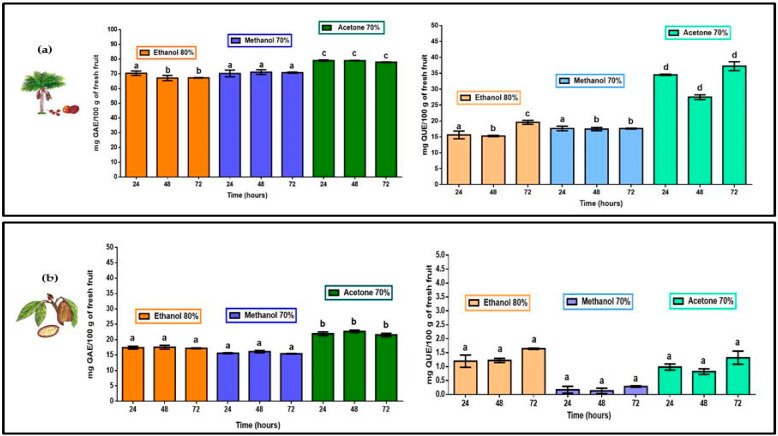
TPC (mg of gallic acid equivalents) and TFC (mg quercetin equivalents) per 100 g of fresh pulp in (**a**) *M. flexuosa* (aguaje) and for (**b**) *T. grandiflorum* (copoazú) extracts in 80% ethanol, 70% methanol and 70% acetone at three different times. Data are means and lower-case letters represent significant differences based on ANOVA followed by Tukey test (*p* < 0.05).

**Figure 3 molecules-26-06431-f003:**
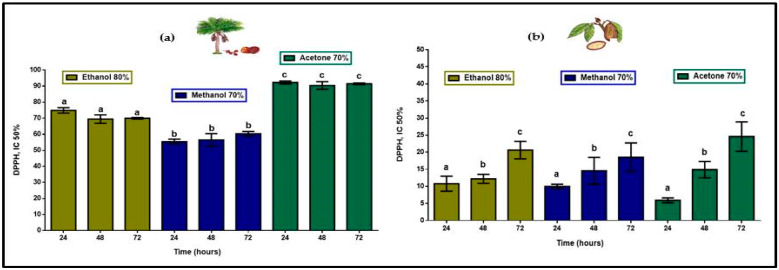
Antioxidant activity (mg AAE/100 g FP) in extracts of (**a**) *M. flexuosa* and (**b**) *T. grandiflorum* in ethanol, methanol, and acetone at 24, 48, and 72 h. Data are means ± standard deviation lower-case letters representing significant differences based on ANOVA followed by a Tukey test (*p* < 0.05).

**Figure 4 molecules-26-06431-f004:**
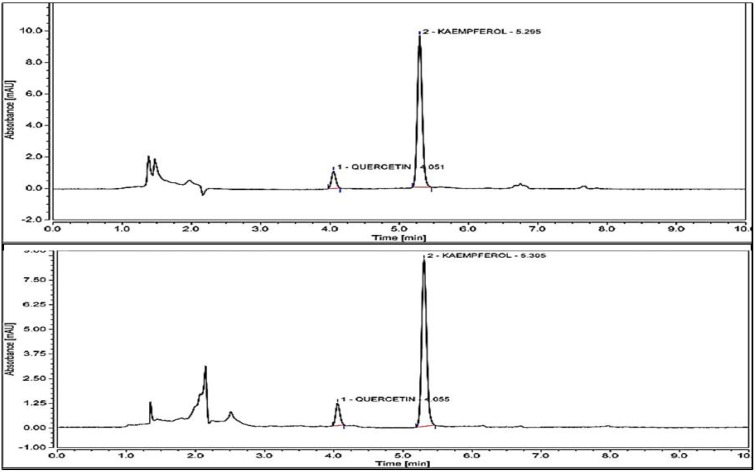
HPLC results for Kaempferol content of *M. flexuosa* 54.43 µg/g extract and *T. grandiflorum* 29.04 µg/g extract. Quercetin in both extracts registered below the limit of quantitation (<LOQ).

**Table 1 molecules-26-06431-t001:** Phenol and flavonoid quantification in fresh pulp of *M. flexuosa* and *T. grandiflorum*.

Fruit Sample	Solvent	TPC * (mg/100 g)	TFC * (mg/100 g)	Flavonoid/Polyphenol Ratio
*M. flexuosa* (aguaje)	Ethanol	204.76 ± 3.43	50.45 ± 0.76	1:4.1
	Methanol	212.28 ± 4.16	52.17 ± 0.62	1:4.0
	Acetone	235.92 ± 0.86	99.23 ±1.55	1:2.4
*T. grandiflorum* (copoazú)	Ethanol	52.05 ± 1.19	4.06 ±0.24	1:12.8
	Methanol	47.04 ± 0.61	0.59 ± 0.22	1:79.9
	Acetone	66.12 ± 1.42	3.13 ± 0.44	1:21.1

Mean values for TPC * (mg of gallic acid equivalents) and TFC * (mg quercetin equivalents) per 100 g of fresh pulp ± standard deviation, *n* = 3. Data represents the addition of concentrations at three different extraction times (24, 48 and 72 h).

## Data Availability

Not available.

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
