# Peer review of "Flavonoid/Polyphenol Ratio in Mauritia flexuosa and Theobroma grandiflorum as an Indicator of Effective Antioxidant Action"

_molecules, 2021, doi:10.3390/molecules26216431_

Round 1

Reviewer 1 Report

The authors have revised the manuscript according to the reviewer's comments. The manuscript now reads more meaningful and provides useful information to the readers. I recommend for publication.

Reviewer 2 Report

Manuscript Molecules-1420077titled “Flavonoid/Polyphenol ratio in Mauritia flexuosa and Theobroma grandiflorum as an indicator of effective antioxidant action” by Juan C. Carmona-Hernandez et al., is a Communication and its aim is to identify the best extraction solvent leading to the highest yield of polyphenols and flavonoids in the flash of two Amazonian fruits. 

This manuscript is the revised version of “Molecules -1120108 by Mai Le et al. “document.

Authors provided all the corrections requested and now the manuscript can be considered for a possible publication.

This manuscript is a resubmission of an earlier submission. The following is a list of the peer review reports and author responses from that submission.

Round 1

Reviewer 1 Report

The study discussed in the article is very routine. The article needs some revisions. The manuscript depicts no novelty. These days I have been reviewing several papers from MDPI (different journals like Molecules, Antioxidants, Food etc.) where fruit/veg extracts have been made and analyzed for phenolic and flavonoid content. This article did the same. However in other articles, the studies are corroborated by some applications e.g. in cell biology (cancer etc.) studies, or the extracts are characterized by NMR/mass spec to find out the exact molecule/compound present in the extract. Hence considering the high quality of the journal Molecule, the article in the present form can't be justified for publication.

Line 94: "aqueous extract is 1.86 folds..." - Detailed description on the aqueous extract is missing. The focus of the article is on ethanol and acetone extract. Suddenly mentioning aqueous extract is not clear.

Line 138: "sodium nitrate (NaSO2)" - Correction needed

Reviewer 2 Report

Manuscript Molecules -1120108 titled “Flavonoid/Polyphenol ratio in Mauritia flexuosa and Theobroma grandiflorum as an indicator of applied and effective flavonoid action” by Mai Le et al., is a research article and its aim is to  identify the best extraction solvent leading to the highest yield of polyphenols and flavonoids in the flash of two Amazonian fruits. 

However, the experimental results show the results of the preliminary experiment. Authors should include other experiments.

The authors use only two extraction solvents. The authors should use other solvents and other percentages of solvents for the extraction in order to claim to have found the "...best extraction solvent leading to the highest yield of polyphenols and flavonoids...”.

In the HPLC profile (Figure 3), the authors have identified only two polyphenol peaks (quercetin and kaempferol).  Joilane Alves Pereira Freire et al. reviewed the phytochemistry profile, nutritional and pharmacological activities of M. flexuosa., and they observed numerous polyphenols (J. Food Sci. 2016. 81(11). doi: 10.1111/1750-3841.13529); this is reported in the reference 24 too.

In figure 3: delete panel a. It is useless to insert the standards profile. It is more important to include the profiles of each extraction performed. Add a more complete HPLC analysis of polyphenolic extract.

Then, the authors declared  "....The most studied biological property in polyphenols is related to oxide-reducing activity...". The authors could perform some in vitro tests (for example DPPH free radical method) to evaluate the antioxidant capacity of the different extracts obtained.